# Research Progress on the Pharmacodynamic Mechanisms of Sini Powder against Depression from the Perspective of the Central Nervous System

**DOI:** 10.3390/medicina59040741

**Published:** 2023-04-10

**Authors:** Zhongqi Shen, Meng Yu, Zhenfei Dong

**Affiliations:** 1College of Traditional Chinese Medicine, Shandong University of Traditional Chinese Medicine, Jinan 250355, China; 2Innovative Institute of Chinese Medicine and Pharmacy, Shandong University of Traditional Chinese Medicine, Jinan 250355, China

**Keywords:** Sini Powder, depression, pharmacodynamic mechanism, central nervous system, blood–brain barrier

## Abstract

Depression is a highly prevalent emotional disorder characterized by persistent low mood, diminished interest, and loss of pleasure. The pathological causes of depression are associated with neuronal atrophy, synaptic loss, and neurotransmitter activity decline in the central nervous system (CNS) resulting from injuries, such as inflammatory responses. In Traditional Chinese Medicine (TCM) theory, patients with depression often exhibit the liver qi stagnation syndrome type. Sini Powder (SNP) is a classic prescription for treating such depression-related syndrome types in China. This study systematically summarized clinical applications and experimental studies of SNP for treatments of depression. We scrutinized the active components of SNP with blood–brain barrier (BBB) permeability and speculated about the corresponding pharmacodynamic pathways relevant to depression treatment through intervening in the CNS. Therefore, this article can enhance our understanding of SNP’s pharmacological mechanisms and formula construction for depression treatment. Moreover, a re-demonstration of this classic TCM prescription in the modern-science language is of great significance for future drug development and research.

## 1. Introduction

Depression is one of the leading risk factors for suicide, affecting more than 300 million people worldwide [1]. The pathological causes of depression can vary, as it is associated with neuronal atrophy, synaptic loss, and a decline in relevant neurotransmitter activity in the prefrontal cortex and hippocampus [2,3,4,5]. Although current anti-depression treatments can be effective, effective action requires up to 6 weeks with multiple therapeutic agents, while side effects are observed [6]. Selective serotonin reuptake inhibitors (SSRIs), a typical type of anti-depression drug via restoring neuroplasticity, may lead to adverse effects, such as headache, insomnia, fatigue, and somnolence, which eventually increase suicide rates in adolescents with depression in the early stages of treatment [7,8,9,10]. Hence, novel therapeutic options are needed for the safe and effective treatment of depression.

The onset and progression of depression are intimately associated with the central nervous system (CNS). For instance, neuron cell damage caused by inflammation and the neurotransmitter dysfunction represented by a decrease in 5-hydroxy tryptamine (5-HT) undoubtedly play considerable roles in the process of depression [11,12]. These findings direct the paths of antidepression drug discovery, namely, that the pharmaceutical components should exhibit the capability to cross the blood–brain barrier (BBB) and interfere with the relevant biological pathways, including combating inflammatory damage, inhibiting neuronal apoptosis, restoring synaptic plasticity, and increasing neurotransmitter activity [13,14].

Traditional Chinese Medicine (TCM) is a time-proven medical discipline that originated from Chinese philosophy and religion and is focused on maintaining or restoring holistic balance in the body [15]. Intrinsically, its efficacy relies on the synergic effects of herbs with empirical evidence [16]. Corresponding with the buildup of scientific perspectives on TCMs, their usages have been extensively implemented in medical management for diseases, including depression, due to their multi-component, -target, and -pathway nature with fewer reported side effects [17]. In TCM theory, patients with depression often express liver qi stagnation. Sini Powder (SNP), a classical prescription in the *Treatise on Febrile Diseases* by Zhang Zhongjing in the Han Dynasty, is the standard prescription used to treat depression for this syndrome type by dispersing and relieving stagnated liver qi [18]. Consistently, SNP has been widely used in the clinic to combat depression with significant efficacy and a low incidence of adverse effects, showing good research value and development prospects.

Network pharmacology, as part of bioinformatics technology, integrates system biology with computational biology [19], revealing the relationship between molecular monomers and specific diseases by comprehensively studying and expanding their putative intersected targets [20]. Herein, we identified the active components of SNP with BBB permeability, along with their relevant enriched biological pathways. Then, we reviewed and summarized the previous studies based on these results, which can reflect the pharmacodynamical mechanisms of SNP in the treatment of depression regarding the CNS. Together, these would not only improve the safety and efficacy of related therapeutic agents but also lay the foundation for future mechanistic studies and anti-depression drug development.

## 2. Sini Powder against Depression

### 2.1. Advantages of Sini Powder against Depression

Chaihu (CH, *Radix Bupleuri*), Zhishi (ZS, *Aurantii Fructus Immaturus*), Baishao (BS, *Paeoniae Radix Alba*), and Gancao (GC, *Licorice*) are included in SNP [21]. Multiple clinical trials demonstrated that SNP can effectively modulate the circadian rhythms of depression patients [22], demonstrating therapeutic effects on several types of depression, including diabetes with depression [23], functional dyspepsia with depression [24], and post-stroke depression [25]. Animal experiments proved that SNP can regulate the hypothalamic–pituitary–adrenal (HPA) axis, and a single pretreatment extract can relieve the increases in serum corticosterone (CORT) and plasma corticotropin-releasing hormone (CRH) levels induced by acute stress, as well as elevating the mRNA expression of the hippocampal glucocorticoid receptor to combat depression [26]. Similar prescriptions of SNP can also interfere with neurogenesis, up-regulate tight BBB connections, and balance the fibrinolytic system to reduce depressive behavior [27]. Experiments in rats have shown that SNP can inhibit the activation of inflammatory responses mediated by the nuclear factor-kappa B (NF-κB) pathway, as well as inflammasomes, such as nod-like receptor protein 3 (NLRP3), interleukin (IL)-1β, and IL-6 [28]. In addition, SNP can also prolong sleep time and alleviate insomnia mood disorder [29].

### 2.2. Therapeutic Effects of Sini Powder Herbs against Depression

The Chaihu and Baishao (CH–BS) drug pair is one of the most accepted combinations among the multiple TCM classical antidepressant prescriptions. Studies have shown that the antidepressant effects of CH–BS are significantly better than those of CH or BS alone in a sucrose preference test (SPT), open field test (OFT), and forced swim test (FST) [30]. The CH–BS drug pair can reduce the generation of peroxidation products and down-regulate NLRP3 inflammasome expression, IL-1β, IL-6, and tumor necrosis factor (TNF)-α secretion to treat depression by combating inflammatory response and oxidative stress damage [31]. A single application of CH also has antidepressant effects, with a meta-analysis on CH proving that the efficacy and safety aspects of combining CH with antidepressants were generally superior to single-antidepressant treatment [32].

## 3. Pharmacodynamic Mechanisms of Active Components of SNP

### 3.1. Screening of Active Components Crossing the BBB

We adopted a network pharmacology approach to confirm the active components that can cross the BBB. The SNP components were collected from the Traditional Chinese Medicine System Pharmacology Database and Analysis Platform (TCMSP, https://old.tcmsp-e.com/tcmsp.php (accessed on 14 December 2022)). Oral bioavailability (OB) is one of the most considerable pharmacokinetics parameters, and the higher the OB value the better the drug-likeness (DL) of the active component [33]. The Caco-2 screening assay is a valuable tool for testing compounds for intestinal permeability [34]. In this study, the criteria of OB ≥ 30%, DL ≥ 0.18, and Caco-2 ≥ −0.4 were used to screen the active components obtained from the TCMSP [35].

The canonical SMILES of all active components obtained from the TCMSP database were imported into the Swiss ADME (http://www.swissadme.ch/ (accessed on 14 December 2022)) website to analyze whether they can cross the BBB. Finally, a total of 40 active SNP components crossing the BBB were screened out, and the results are listed in Table 1.

### 3.2. Active Components of Zhishi

Sinensetin (MOL001803) can inhibit the activation of the toll-like receptor 4 (TLR4) and NF-κB pathways; significantly inhibit the production of pro-inflammatory mediators, such as TNF-α and IL-6; promote the activation of anti-inflammatory cytokines, such as IL-4, IL-10, IL-13, and TGF-β; and also reduce the production of MMP9 and MMP13 [36,37], thereby protecting the degraded extracellular matrix and attenuating cell apoptosis [38]. Additionally, it can decrease apoptosis by elevating the phosphorylation of phosphatidylinositol 3-kinase (PI3K) and protein kinase B (Akt) and activating the PI3K-Akt pathway [39]. Sinensetin also attenuates the activation of the mitogen-activated protein kinase (MAPK) pathways in brain microvascular endothelial cells by decreasing the phosphorylation of extracellular signal-regulated kinase (ERK), c-Jun N-terminal kinase (JNK), and p38 subclasses, and can up-regulate peroxisome-proliferators-activated receptor gamma (PPARG) system activity, thereby restoring neuroplasticity and neurogenesis [40,41].

Didymin (MOL005849) inhibits the NF-κB pathway through the MAPK pathway, decreases NLRP3 inflammasome levels, and down-regulates the expression of inflammatory factors, such as IL-1β, IL-6, and TNF-α, as well as monocyte chemotactic protein 1 (MCP-1) [42]. In vivo and vitro experiments have proven that its use in pretreatment can attenuate apoptosis resulting from inflammatory response damage through decreasing Bax and Caspase-3 levels and increasing the Bcl-2 level [43]. Didymin can also activate the PPAR signaling pathway, increase the expression of PPARG to protect neuronal cells, and activate the PI3K-Akt-GSK-3β pathway to repair synaptic plasticity [44,45].

Tetramethoxyluteolin (MOL007879) can inhibit the focal inflammation in the brain caused by the cytokine of mast cells and microglia, which results from the co-stimulation of CRH in cooperation with IL-33, to exert neuroprotective effects [46]. Isosinensetin (MOL013277) also inhibits the NF-κB pathway through the MAPK pathway to attenuate inflammatory damage [47].

### 3.3. Active Components of Baishao

Palbinone (MOL001919) can significantly reduce the production of pro-inflammatory cytokines IL-18 and IL-1 and improve the accumulation of nuclear factor erythroid-2-related factor 2 (Nrf2), thereby modulating inflammation-induced, depressive-like responses through the Nrf2-NLRP3 pathway [48].

Paeoniflorin (MOL001925) significantly inhibits the secretion of inflammatory factors, such as IL-6ˏ TNF-α [49]. In vitro and vivo studies demonstrated that paeoniflorin can decrease Bax and Caspase-3 expression levels, increase the Bcl-2 expression level, and inhibit neuronal apoptosis by activating the PPARG system [50,51]. Paeoniflorin can also increase 5-HT content in the serum of depression model rats to restore neurotransmitter function [52]. Furthermore, the data have indicated that paeoniflorin can effectively improve the depression-like state in depression model rats, the behavioral profile and HPA axis can be significantly regulated in the pharmacodynamic index, and the metabolic pathways of the related indexes can be significantly recovered in the metabolomics analysis—with the mechanism possibly related to reductions in the serum levels of ACTH, CORT, and CRH [53,54]—to intervene in the neuroendocrine immune system and metabolic pathways [55]. In addition, paeoniflorin can activate the downstream Nrf2 pathway through the PI3K-Akt pathway to combat oxidative stress [56,57]. Additionally, paeoniflorin can also reduce the protein expression of ERK, JNK, and p38, and the potential mechanism may be related to the inhibition of the TGF-β-activated kinase 1 (TAK1)-MAPK signaling pathway [58,59].

### 3.4. Active Components of Gancao

Formononetin (MOL000392) attenuates the over-production of cytokines and inflammatory mediators; decreases the serum levels of inflammatory factors, such as TNF-α and IL-6; and inhibits the IL-1β-induced activation of the NF-κB pathway [60]. It also inhibits the expression of cyclooxygenase 2 (COX-2) and inducible nitric oxide synthase (iNOS), as well as the synthesis of catabolic factors, such as matrix metalloproteinases (MMPs) [61], to combat the damage to cells caused by inflammation and oxidative stress. Formononetin could also interfere with the expression level of the 5-HT receptor, modulating 5-HT activity [62]. Formononetin can up-regulate the PI3K-Akt pathway by activating the downstream regulator of the phosphatase and tensin homolog (PTEN) to improve neuronal atrophy [63,64] and depressive behavior in the mouse model of depression by inhibiting the activity of PI3K-Akt-GSK-3β and the downstream Notch1 signaling pathway [65]. Furthermore, formononetin also exerts anti-neurodegenerative effects by inhibiting apoptosis through decreasing MAPK phosphorylation [66].

Licochalcone A (MOL000497) potently inhibits the release of inflammatory factors TNF-α, IL-4, IL-5, IL-6, and IL-13 [67,68]. It inhibits the activity of NLRP3 inflammatory bodies by reducing the production of IL-1β, directly binds to the accessory proteins of TLR4, and inhibits TLR4 pathway activation and inflammatory cytokine induction, significantly reducing immune cell infiltration to exert anti-inflammatory and neuroprotective effects [69,70]. Licochalcone A could interfere with the 5-HT receptor expression level, modulating 5-HT activity [71], as well as intervene in the Akt and NF-κB pathways by down-regulating MAPK expression, which inhibits the cell migration and invasion resulting from inflammatory response [72,73]. It can also up-regulate PPARG protein expression [74].

Treatment with vestitol (MOL000500) can reduce the activity of NF-κB and ERK pathways, decrease the levels of pro-inflammatory factor IL-6, IL-4, TNF-α, TGF-β, and NO synthase, promote the release of anti-inflammatory factor IL-10 to combat oxidative stress and inflammatory damage, and inhibit leukocyte migration [75]. Maackiain (MOL001484) can reduce the secretion of MCP-1 and TNF-α, as well as inhibit the NF-κB pathway. In parallel, it can also up-regulate the expression of Bcl-2; down-regulate the expression of Bax, Caspase-9, and Caspase-3; and inhibit the apoptosis resulting from inflammatory responses [76].

Liquiritigenin (MOL001792) has been shown to have anti-inflammatory, anti-hyperlipidemic, and antioxidant properties, which can inhibit the activation of the NF-κB and NLRP3 inflammatory pathways [77]. It has also been shown to have ameliorative effects on the HPA axis [78]. Experiments demonstrated that liquiritigenin can reduce the depressive behavior of mice through its anti-inflammatory and neuroprotective effects, and it may also treat chronic depression through the brain-derived neurotrophic factor (BDNF) signaling pathway mediated by the PI3K-Akt-mTOR signaling pathway [78,79,80].

Medicarpin (MOL002565) was experimentally demonstrated to be able to inhibit CUMS-induced neuro-inflammation through hampering NF-κB pathway activation [81], and inhibit brain microvascular endothelial cell injury by activating the PI3K-Akt-FOXO pathway [82]. Shinpterocarpin (MOL004891) was reported to have antioxidant, anti-inflammatory, and anti-cytotoxic activities, as well as exhibiting significant binding activity against the PPARG ligand [83,84].

Glabridin (MOL004908) can significantly down-regulate the expression of pro-inflammatory mediators, such as IL-1β and TNF-α; prevent the decline in Nrf2 activity and inhibit TLR4 activation; and up-regulate Bax expression, reversing inflammation-induced apoptosis [85,86]. Additionally, it can inhibit the NF-κB pathway through the MAPK pathway and activate the PI3K-Akt-GSK-3β pathway to repair synaptic plasticity [87,88]. Glabridin can also suppress inflammation and oxidative stress injury by activating the PPARG system, inhibiting pro-inflammatory factor expression, and elevating anti-inflammatory factor expression, thus reducing the level of peroxidation products and increasing the level of peroxidases [89].

## 4. Pharmacodynamic Pathways of Active Components of SNP

To obtain the pharmacodynamic pathways of SNP in CNS treatments, we speculated on the potential targets of active components and applied enrichment analysis to obtain the pharmacodynamic pathways that the potential targets may act on. After entering into the CNS, the active components should be able to treat depression effectively by intervening in the relevant pathways. 

### 4.1. Acquisition of Potential Targets against Depression

Depression targets were collected from seven databases, including DisGeNET, GeneCards, NCBI, OMIM, UniProt, TTD, and DrugBank, with “depression” as the keyword, while 7948 depression targets were obtained after de-duplicating the results. In total, 695 active-component targets that can cross the BBB were found by importing the canonical SMILES into the SwissTargetPrediction website. In total, 559 targets that intersected depression targets and active components were considered the potential SNP targets against CNS depression.

### 4.2. Enrichment Analyses

The efficacy of TCM is represented by the synergy effects of potential targets; however, experimental validation would require additional work, thus hindering our progression with scrutinizing the TCM target–effects relationship spectrum. KEGG enrichment analysis shows considerable numbers of signal pathways for the synergy effects of potential targets. Therefore, we need to step-wisely narrow our study objects through KEGG analysis. The MCODE algorithm can detect densely connected regions that are likely to represent molecular complexes in large PPI networks based solely on connectivity data [90]. Both are common approaches to studying the relationship between TCM prescriptions and their efficacy against diseases.

The Metascape online analysis website was used for the Kyoto Encyclopedia of Genes and Genomes (KEGG) pathway enrichment analyses. The top 20 KEGG pathways of SNP were sorted by their *p*-values (shown in Figure 1A), and the results were mainly concerned with the neuroactive ligand–receptor interaction, calcium signaling pathway, and MAPK signaling pathway. Given the highly distracting capacity of cancer pathways, relevant cancer pathways were excluded, accordingly. For the MCODE analyses, the active components that can cross the BBB and potential targets were imported into Cytoscape 3.9.1 software (Boston, MA, USA), and the MCODE plug-in was used to identify the top three clusters, which were screened out (Figure 1B). The details of each cluster are listed in Table 2, and the targets of each cluster are listed in Appendix A. According to the results, the clusters were related to apoptosis, the cell cycle, and the PI3K-Akt signaling pathway.

## 5. Discussion and Limitations

Neuroinflammation and its damage are well-known to play a considerable role in the pathological progression of depression. Additionally, the effects of the SNP active components that we reviewed on the CNS are also mostly associated with anti-inflammation and the repair of inflammatory damage. It is worth mentioning that the pharmacological pathway results obtained through enrichment analysis also echoed the review content in the previous text. It is also of concern that the NF-κB pathway, the classical inflammatory signaling pathway, accounted for a significant part of the treatment of depression using SNP. Although the results of the enrichment analysis were not directly linked to the NF-κB pathway, other pathways closely related to it were oriented. Detailed information is discussed below.

According to the KEGG pathway enrichment analysis results, neuroactive ligand–receptor interactions are associated with the transmission of neurotransmitters across synapses, represented by 5-HT [91,92]. The BDNF-related pathways can influence 5-HT activity by acting on the expression of its receptor to restore normal neurotransmitter function [93,94,95]. The PPARG system can rapidly sense cellular stress responses and protect glial cells, neuronal cells, and cerebrovascular endothelial cells in the CNS, as well as considerably increasing BDNF levels in brain regions to promote neuroplasticity and neurogenesis [96,97]. The HPA axis has been implicated in the pathophysiology of multiple emotional and cognitive disorders by data from different neuroscience disciplines [98,99], including correcting the abnormal secretion of adrenocorticotropic hormone (ACTH) [100,101], CORT [102,103], and CRH [104]. Additionally, the hypersecretion of hormones can lead to the impairment of 5-HT receptor activity [105]. Studies have shown that stress plays a key role in the pathogenesis of depression, and the stress-induced MAPK pathway includes the ERK, JNK, and p38 subclasses. ERK expression would be involved in the intermediate process of BDNF expression, and the inhibition of ERK in the prefrontal cortex and hippocampus leads to depression-like behavior [106,107]. Abnormal phosphorylation of JNK can promote apoptosis and inhibit neural synapse regeneration, leading to depression [108]. Additionally, the activation of p38 MAPK is closely related to stress-induced aversive responses [109]. 

Additionally, regarding the MCODE results, stress is well-known to exert deleterious effects on neuronal apoptosis and neuroplasticity by affecting multiple cellular cascades, which can also lead to structural plasticity changes in the brain [110]. The regulation of the cell cycle is essential for apoptosis [111]. Neuronal apoptosis decline and the enhancement of neuronal proliferation after cell cycle regulation have proven to be effective in treating depression [112]. In addition, there are multiple factors responsible for the occurrence of depression, such as damage due to neuroinflammation. The NF-κB pathway leads to neuroinflammation, and most anti-inflammation pharmacodynamic pathways are related to it [113,114]. Activation of the PI3K-Akt cell survival pathway can inhibit the downstream NF-κB pathway [115], inhibit the forkhead box protein O 1 (FoxO1) pathway to attenuate the apoptosis resulting from neuro-inflammation [116], and inhibit the glycogen synthase kinase-3β (GSK-3β) pathway to restore synaptic plasticity [117,118]. The PI3K-Akt and MAPK pathways are closely linked and, together, modulate neuronal cell survival [119,120].

Despite encouraging findings, certain limitations remain. First, there have been few clinical trials targeting SNP for the treatment of depression, especially those related to the CNS. The reason might be the poor BBB penetration rate. We expect more clinical trials regarding SNP treatment for depression to emerge in follow-up studies, preferably in relation to the active components that we have screened out that can enter into the CNS. Second, our study was mainly based on the current databases, and given that the number of TCM studies on depression is still insufficient, it is inevitable that the absence of herbal components in the SNP may have an impact on the pharmacodynamic mechanism results of the enrichment analysis. Finally, although we have speculated on the active components in SNP that can enter the CNS, further validation experiments are still required, such as the HPLC detection of active components using rat cerebrospinal fluid extracted after SNP treatment.

## 6. Conclusions and Prospects

Because the pathological process of depression in the CNS is incompletely understood, the development of novel medicines remains a serious challenge. SNP, a classic TCM prescription, is widely utilized to combat depression in clinical applications with obvious efficacy and fewer reports of side effects. In this study, we obtained active SNP components that can intervene in the CNS, and systematically reviewed and speculated on their antidepressant pharmacodynamic mechanisms. It is quite evident that most of the active components have been confirmed to possess significant anti-inflammatory properties. Given that inflammatory response occupies a very considerable position in the pathological process of depression and has a close association with other etiological factors, we believe that exploring the anti-inflammatory properties of these active components will be a powerful direction for SNP follow-up studies to take in determining its efficacy as an antidepressant.

Advances in modern biological technology will keep accelerating the research on SNP’s mechanisms against depression, improving the safety and efficacy of the related therapeutic agents. After further identifying the effective components in SNP that can intervene in the CNS through molecular docking and molecular dynamics simulation, in the future, we will encourage further attention to improving the efficiency with which these components enter the CNS, such as through nanomaterial wrapping transport, thereby increasing the efficacy of SNP as an antidepressant. In addition, more quantitative detection of the active components through technical means, such as network pharmacology and HPLC, which can more rigorously determine the standard composition of SNP in clinical applications and experimental studies, will benefit the subsequent research on, and industrialization of, SNP. Moreover, TCM prescriptions usually have characteristics of being homologous to food, which can benefit the health industry to some extent through the confirmation and extraction of the pharmacodynamic components of SNP and its subsequent application in combination with diet.

## Figures and Tables

**Figure 1 medicina-59-00741-f001:**
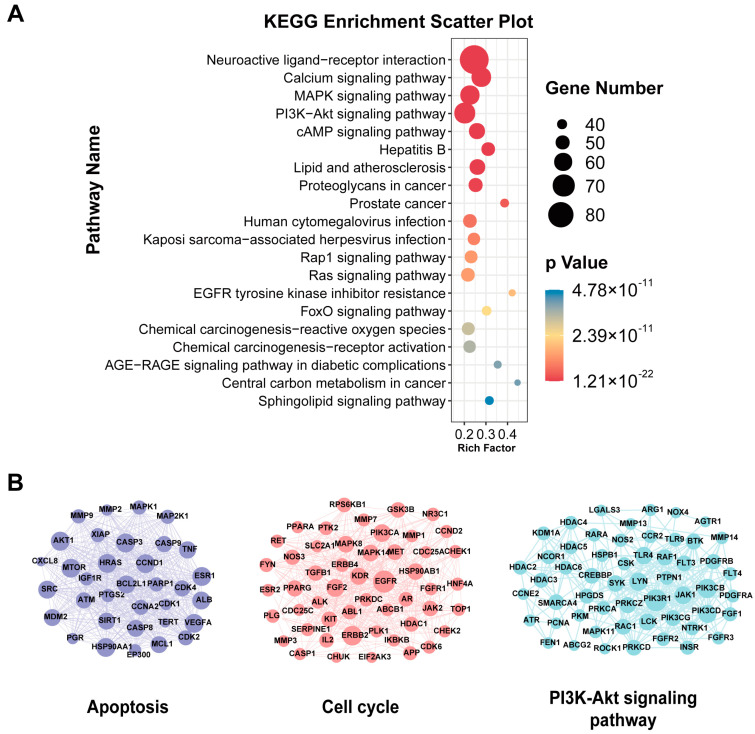
Enrichment analyses. (**A**) KEGG pathway enrichment analyses of SNP against depression, sorted by *p*-value. (**B**) The top 3 clusters of SNP, as identified using the MCODE algorithm.

**Table 1 medicina-59-00741-t001:** Active components crossing the BBB of all 4 herbs of SNP.

Herb	MOL ID	Active Component
CH	MOL004644	Sainfuran
	MOL013187	Cubebin
ZS	MOL001803	Sinensetin
	MOL001941	Ammidin
	MOL005849	Didymin
	MOL007879	Tetramethoxyluteolin
	MOL013277	Isosinensetin
	MOL013279	5,7,4’-Trimethylapigenin
	MOL013430	Prangenin
	MOL013435	Poncimarin
	MOL013436	Isoponcimarin
	MOL013437	6-Methoxy aurapten
BS	MOL001919	Palbinone
	MOL001925	Paeoniflorin
GC	MOL000392	Formononetin
	MOL000497	Licochalcone a
	MOL000500	Vestitol
	MOL001484	Maackiain
	MOL001792	Liquiritigenin
	MOL002565	Medicarpin
	MOL003896	7-Methoxy-2-methyl isoflavone
	MOL004815	Kanzonol B
	MOL004833	Phaseolinisoflavan
	MOL004835	Glypallichalcone
	MOL004838	Kanzonol U
	MOL004891	Shinpterocarpin
	MOL004908	Glabridin
	MOL004910	Glabranin
	MOL004911	Glabrene
	MOL004941	(2R)-7-hydroxy-2-(4-hydroxyphenyl)chroman-4-one
	MOL004945	Isobavachin
	MOL004957	Isoformononetin
	MOL004959	1-Methoxyphaseollidin
	MOL004966	3’-Hydroxy-4’-O-Methylglabridin
	MOL004974	3’-Methoxyglabridin
	MOL004978	4’-Methoxyglabridin
	MOL004980	Inflacoumarin A
	MOL004988	Kanzonol F
	MOL004991	7-Acetoxy-2-methylisoflavone

**Table 2 medicina-59-00741-t002:** Top 3 clusters identified from the MCODE results of SNP.

Cluster	Pathway Description	Number of Targets	Number of Edges	Score
1	Apoptosis	34	502	30.424
2	Cell cycle	51	415	16.600
3	PI3K-Akt signaling pathway	55	283	10.481

## Data Availability

The data used to support the findings of this study are included within the article and Appendix A.

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
