# Peer review of "Research Progress on the Pharmacodynamic Mechanisms of Sini Powder against Depression from the Perspective of the Central Nervous System"

_medicina, 2023, doi:10.3390/medicina59040741_

Round 1

Reviewer 1 Report

1.       In the abstract it is stated: “This review aims to promote” It is better to state the "objective" after the introduction and then report the findings and conclusions

2.       Please in introduction section, explain the scientific background and rationale exactly not just by linking variables.

3.       In methods, describe the periods of recruitment

4.       In methods, please describe any efforts to address potential sources of bias.

5.       In discussion, please discuss limitations of the study, taking into account sources of potential bias or imprecision. Also, discuss both direction and magnitude of any potential bias.

6.       In the Limitations section, point out the factors that have restricted your research's internal and external credibility and state the methodological limitations. Make research suggestions based on these limitations and write practical recommendations based on the findings. Avoid making general suggestions, such as holding and explaining the findings based on the hypothesis's results.

7.       We consider recommendations for future research an essential aspect of the process by which reviews develop thinking. This article should therefore be diligent in generating new questions that need to be addressed through future empirical research and serve to drive the field forward by posing the questions and hypotheses that need to be the targets of new studies.

Reviewer 2 Report

Although the manuscript seems interesting, there are some major issues that must be resolved.

1-      The main effects of most of the components of SNP target the inflammatory pathways, the authors should emphasis the role of inflammation in depression and its effect on the neurotransmitters.

2-      The aim of the study is not clear.

3-      In the Enrichment Analyses the authors reported that the cAMP-response element binding protein (CREB) and brain-derived neurotrophic factor (BDNF) pathway can influence 5-HT activity by acting on expression of its receptor, to restore normal neurotransmitter function, without any evidence for the effect of any components of SNP on the CREB or BDNF.

4-      This is a review not original research; it should systematically review the results of other published studies do not include a new docking or new works.

5-      The authors used KEGG pathway enrichment analysis and MCODE algorithm to predict the pathways of SNP through intervening in the CNS to treat depression. Does it enough for proving it efficacy on depression?

6-      All figures’ resolutions need to be improved.

Reviewer 3 Report

The study shows interesting results but it is difficult to follow. The organization is confusing. The authors should reorganize the manuscript as an article instead of a review so after the introduction, they should describe their results and then, they should discuss their results, including the links with previous findings. In addition to the required reorganization which is my main concern, other minor concerns are:

-       The font in the figures is too small and difficult to read, please make it bigger if possible.

-       Depression in the abstract and the introduction must not be defined with the same word, I mean, “Depression is…… characterized by persistent depression” sounds redundant.

Round 2

Reviewer 2 Report

The authors have addressed my comments

Author Response

We have re-edited the English language of our manuscript through MDPI (English-64131).

Reviewer 3 Report

The authors addressed my concerns

Author Response

(The authors gave the same response as above.)
